# Nitrogen nutrition contributes to plant fertility by affecting meiosis initiation

Han Yang [1,2,4], Yafei Li [1,2,4], Yiwei Cao [1,2], Wenqing Shi [1], En Xie[1,2], Na Mu [1,2], Guijie Du[1], Yi Shen [1], Ding Tang [1] & Zhukuan Cheng [1,2,3 ✉]

Nitrogen (N), one of the most important plant nutrients, plays crucial roles in multiple plant developmental processes. Spikelets are the primary sink tissues during reproductive growth, and N deficiency can cause floral abortion. However, the roles of N nutrition in meiosis, the crucial step in plant sexual reproduction, are poorly understood. Here, we identified an N-dependent meiotic entrance mutant with loss of function of *ELECTRON TRANSFER FLA-VOPROTEIN SUBUNIT β* (*ETFβ*) in rice (*Oryza sativa*). *etfβ* displayed meiosis initiation defects, excessive accumulation of branched-chain amino acids (BCAAs) and decrease in total N contents in spikelets under N starvation, which were rescued by applying excess exogenous inorganic N. Under N starvation, ETFβ, through its involvement in BCAA catabolism, promotes N reutilization and contributes to meeting N demands of spikelets, highlighting the impact of N nutrition on meiosis initiation. We conclude that N nutrition contributes to plant fertility by affecting meiosis initiation.

[1] State Key Lab of Plant Genomics, Institute of Genetics and Developmental Biology, Innovation Academy for Seed Design, Chinese Academy of Sciences, 100101 Beijing, China. [2] University of Chinese Academy of Sciences, 100049 Beijing, China. [3] Jiangsu Co-Innovation Center for Modern Production Technology of Grain Crops, Yangzhou University, 225009 Yangzhou, China. [4]These authors contributed equally: Han Yang, Yafei Li. ✉email: zkcheng@genetics.ac.cn

Nitrogen (N), quantitatively the most important plant nutrient, plays crucial roles in multiple developmental processes[1,2]. Under N stress conditions, efficient mechanisms for N remobilization between source and sink tissues are required for enhancing N use efficiency, ensuring sustainable homeostasis and better growth[3–7]. The development of spikelets, which are the primary sink tissues during reproduction, is tightly linked to nutrient availability[6,8]. Many N utilization (or reutilization)-related genes have been identified to be involved in floral development[9–11] and fertility[12,13]. These observations highlight the crucial role of N in plant reproduction.

A criticaley step in sexual reproduction is meiosis, which produces haploid cells from a diploid cell via two successive cell divisions[14]. As the primary determinant of meiosis, meiotic fate acquisition is a complex and precisely regulated process[15], which is cooperatively controlled by the coordinated effects of environmental cues[16–18] and the genetic makeup of the cell[19,20]. It seems that there is a close association between nutrient status and meiosis initiation[17,21]. N stress in budding yeast (Saccharomyces cerevisiae)[22,23] and retinoic acid (RA) in mice (Mus musculus)[24,25] are known to be crucially dependent on meiosis initiation. Although several genes have been characterized to be involved in meiotic entrance in plants[26–28], the effects of nutritional cues on meiotic entrance[14,15], representing a significant gap in understanding this fundamental biological process, which is essential for plant sexual reproduction.

When carbohydrate substrates for respiration are limiting under environmental or developmental stress conditions, an alternative metabolic pathway widely present in plants and animals, the electron transfer flavoprotein (ETF)/electron transfer flavoprotein quinone oxidoreductase (ETFQO) system[29], is involved in protein and lipid catabolism and provides an alternative substrate to feed electrons into mitochondrial electron transport chain[30–32]. ETF is a heterodimer composed of two subunits, α and β, and serves as an obligatory electron acceptor for mitochondrial matrix flavoprotein dehydrogenases, including isovalyl coenzyme A dehydrogenase (IVDH) and D-2-hydroxyglutarate dehydrogenase (D2HGDH)[33–35]. Reduced ETF is re-oxidized by ETFQO, which delivers electrons to the main respiratory chain via ubiquinone reduction[36–38]. In humans (Homo sapiens), defects in ETF result in glutaric acidemia type II (GA II), a fatal disease characterized by accumulation of organic acids in blood and urine resulting from inability to catabolize various acylcoenzymes (acyl CoAs)[39,40]. In Arabidopsis, etfβ exhibits accelerated senescence and early death during extended darkness with significant accumulation of several amino acids and acyl CoAs[41,42]. These emphasize essential roles of ETFβ in the catabolism of branched-chain amino acids (BCAAs) and organic acids in humans and Arabidopsis[30,43].

In this study, we identify the unique homolog of ETFβ in rice through characterization of a mutant with N nutrition-dependent defects in meiotic initiation. Our data reveal a vital function for ETFβ in affecting meiotic initiation via promoting N reutilization in rice under low N stress. And the ETF/ETFQO system is upregulated under N stress conditions, which can be a safeguard mechanism for plants to maintain fertility in barren soil. Taken together, our results suggest that N nutrition contributes to plant fertility by affecting meiosis initiation, revealing a critical aspect of genotype–environmental nutrition interactions during gametogenesis.

## Results

### Characterization of a sterile rice mutant with defects in pollen mother cell (PMC) differentiation.
To dissect the molecular mechanism of rice meiosis initiation, we performed a large-scale screen for sterile mutants with defects in PMC differentiation using a $^{60}$Co γ-ray radiation-mutagenized mutant library in the indica rice Zhongxian 3037 background. A mutant with obvious sterility was identified (Fig. 1a). Phenotypic characterization of the homozygous lines did not reveal any distinguishing traits compared with the wild type (WT) during vegetative growth.

To clone the causal gene, we crossed heterozygous mutant plants with Zhonghua 11 (a japonica rice variety) and obtained 97 F2 plants for first-pass mapping. The locus was narrowed down to a 2.33-Mb interval on the short arm of chromosome 4. Next, 214 F3 and 507 F4 plants were used for fine mapping, and the linkage region was refined to 170 kb (Fig. 1b). There are 14 genes in this region, almost half of which are small putative expressed proteins or retrotransposon proteins (Rice Genome Annotation Project). Among the remaining genes, LOC_Os04g10400 contained a large fragment deletion (1922 bp) including the 5′-untranslated region (UTR) and the first exon (Supplementary Fig. 1a, b). There were no mutations in other genes within the linkage region. Thus, we regarded LOC_Os04g10400 as the candidate gene. Genetic complementation experiments further confirmed that mutation of LOC_Os04g10400 caused sterility (Supplementary Fig. 2a, b). A plasmid containing the entire open-reading frame (ORF), 5.0-kb upstream region, and 1.8-kb downstream region, rescued the mutant. The deletion mutation caused a substantial reduction of its transcript abundance; transcripts in the mutant were almost undetectable (Supplementary Fig. 2c), representing a loss-of-function allele, consequently.

The male-sterile phenotype of this mutant, which was cultivated in Lingshui, Hainan Province, was characterized by staining transverse anther sections with toluidine blue (TBO) (Fig. 1c). Unlike WT anthers from spikelets of 6–8 mm in length, which were filled with mature pollen, there was no pollen in the mutant spikelets of the same size. Even normal-looking PMCs were rarely observed in the mutant, many smaller cells were gathered in the center of mutant anther locules and surrounded by four or five layers of irregularly arranged somatic cells without tapetum differentiation. Moreover, a large number of cells in the chamber center were apoptotic, resulting in the formation of a cavity (marked with black arrows, Fig. 1c). This phenotype is similar to that of meiosis initiation mutants[26–28].

### Rice ETFβ is the unique homolog of human ETF subunit β.
The causal gene LOC_Os04g10400 encodes the ETF subunit β (ETFβ), which is the only ETFβ homolog in rice. ETFβ is a mitochondrial localized protein (Fig. 1d), containing a single ETF domain (amino acids 23–215) (Supplementary Fig. 3). Alignment of the amino acid sequence of rice ETFβ to the sequences of homologs from bacteria to humans revealed high overall similarity. ETFβ is an evolutionarily conserved protein (Supplementary Fig. 4).

RNA in situ hybridization analysis of transverse sections was performed to examine the localization of ETFβ transcripts in panicles in detail (Fig. 1e). ETFβ was conspicuously expressed in archesporial cells (ARs) located in the four corners of young anthers. As ARs developed into primary sporogenous cells (PSCs) and primary parietal cells (PPCs), the accumulation of ETFβ mRNA obviously increased in PSCs. After the differentiation of four somatic cell layers, ETFβ was preferentially expressed in PMCs, and weaker expression was observed in the tapetum. Thereafter, expression of ETFβ in the anther lobes decreased substantially as the microsporocytes underwent meiosis. The expression pattern of ETFβ is consistent with the defective meiosis phenotype in etfβ, implying that ETFβ may have a critical function in early anther development and meiotic processes.

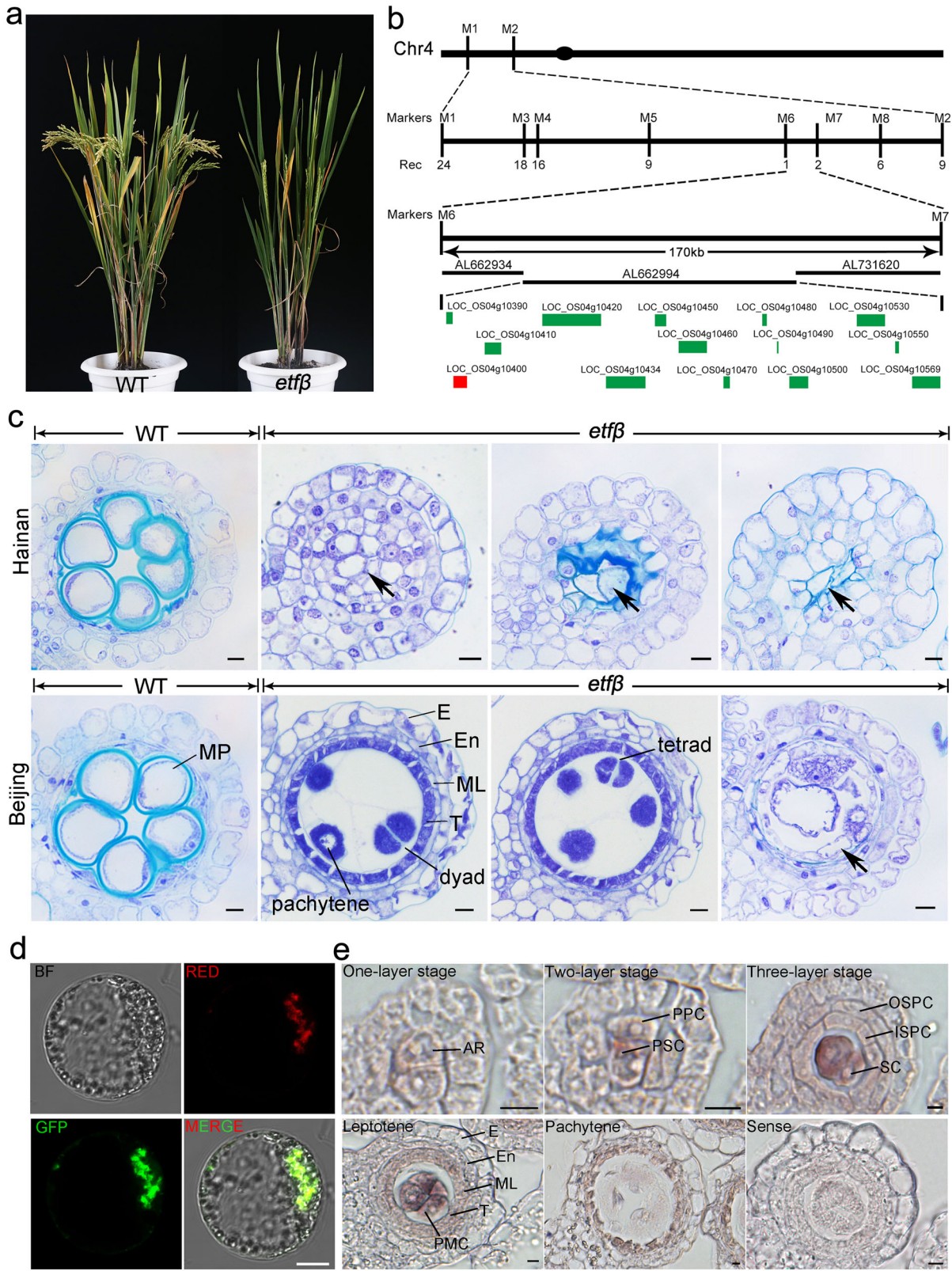

**The severity of meiosis defects in *etfβ* is affected by nutritional status in the cultivation environment**. The phenotype of *etfβ* cultivated in Beijing was obviously different from that of *etfβ* cultivated in Hainan (Fig. 1c). Confusingly, the meiotic progression in spikelets of the same size as WT spikelets (6–8 mm) from *etfβ* grown in Beijing was so random that PMCs from the same

anther chamber arrested at different stages of meiosis, including seemingly normal tetrad, dyad, and pachytene. However, abnormal apoptosis (marked with a black arrow in Fig. 1c) was occasionally observed, and far less common than that in Hainan. *etfβ* showed more severe abnormalities as grown in Hainan than in Beijing.

**Fig. 1 *etfβ* is a sterile mutant with a defective meiosis phenotype when grown in a paddy field. a** Whole plants of wild type (WT) and *etfβ* at heading stage grown in a paddy field. **b** Fine mapping of *ETFβ*. Vertical lines, marker loci; Rec number of recombinant plants. **c** Phenotypic comparison of anthers of the same size between WT and *etfβ* in Hainan (top) and Beijing (bottom); anthers were stained with toluidine blue (TBO). The right three images of *etfβ* in in Hainan and Beijing were aligned in order of the phenotypic severity. The arrows indicate abnormal apoptosis in the center of locules. E epidermis, En endothecium, ML middle layer, MP mature pollen, PMC pollen mother cell, T tapetum. Scale bars, 5 μm. **d** Mitochondrial localization of ETFβ in rice protoplasts. Red indicates mitochondria, and green indicates transient expression of ETFβ fused to green fluorescent protein (GFP) in rice protoplasts. The yellow fluorescence in the merged image (bottom right) shows mitochondrial localization. Scale bars, 10 μm. **e** Expression of *ETFβ* in WT anthers determined by RNA in situ hybridization. Hybridization with the sense *ETFβ* transcript was performed as a negative control. AR archesporial cell, ISPC inner secondary parietal cell, OSPC outer secondary parietal cell; PPC primary parietal cell, PSC primary sporogenous cell, SC sporogenous cell. Scale bars, 5 μm.

Given the variable phenotypic severity of *etfβ*, with more severe abnormalities observed in Hainan than in Beijing, we hypothesized that the defective meiosis might be closely related to fluctuating environmental factors, such as photoperiod, temperature, humidity, and nutrient content in soil. Because of the known association of ETFβ with GA II in humans[34], the role of *Arabidopsis* ETFβ in response to carbon starvation treatment and the biochemical function ETFβ of in catabolizing BCAAs (leucine [Leu], isoleucine [Ile], and valine [Val])[41], combined with the knowledge that ETF/ETFQO system acts as an alternative metabolic pathway under stress conditions[29], we were more inclined to consider nutrient content in soil as the environmental factor responsible for the variation in phenotypic severity of *etfβ*. It is well known that nutrients from the surrounding environment are vital for the survival and growth of all organisms, especially for immobile plants, which cannot migrate to escape inhospitable environments[44,45]. Therefore, we treated *etfβ* with different concentrations of whole nutrients using an irrigation control system with soilless culture nutrient liquid containing all the inorganic elements needed for rice growth and development (Supplementary Fig. 5). Three concentration gradients were set: twice the normal concentration of Hoagland solution (2×), the normal concentration of Hoagland solution (1×), and double distilled water instead of Hoagland solution (0×). Among them, 2× was used to supply a eutrophic environment, and conversely, 0× represented an extreme inalimental condition, while 1× was conducted as a positive control.

It is worth mentioning that the phenotypes of *etfβ* exhibited perfect correlation with nutrient condition (Fig. 2a–d). The anthers of *etfβ* treated with 2× solution were golden yellow with a normal size observed under a stereoscope. Moreover, we observed plump pollen grains filling the four chambers via $I_2$-KI staining and transverse sections of anthers stained with TBO. It seems that excess N enabled *etfβ* to complete meiosis and anther development and produce viable microspores, and there was no significant difference between *etfβ* and WT (Fig. 2a, d). In contrast, 0× solution induced the most severe defects in *etfβ*, whose anthers were smaller compared with those under the 2× and 1× conditions, did not undergo dehiscence, and were apparently devoid of pollen grains (Fig. 2b–d). Cells in the center of the chamber were abnormally apoptotic, and only the epidermis remained, forming shriveled chambers (Fig. 2b). In addition, under the 1× condition, we detected variable defects in *etfβ* anthers as mentioned above, including different degrees of stagnation and apoptosis. Occasionally, a few pollen grains could be observed in *etfβ* under the 1× condition (Fig. 2c). The phenotype of *etfβ* was clearly opposite under the two extreme nutritional conditions we applied, the 2× and 0× conditions. The meiotic defects observed in *etfβ* were relieved as excess nutrients were added. The phenotypes in *etfβ* anther locules changed regularly with the nutrient concentrations: the anthers in *etfβ* changed from four shriveled chambers under the 0× condition, to partial (two or three) locules with pollen grains under the 1× condition, and four locules full of mature pollen under the 2×

condition (Fig. 2e). We conclude that it is the different nutritional conditions that cause varying degrees of meiotic defects in *etfβ*.

**Both male and female meiosis progression in *etfβ* were affected by N status and were largely dependent on excess exogenous inorganic N.** We demonstrated that the phenotype of *etfβ* can be controlled by changing the nutrient concentration. However, it was yet unclear which element or compound was responsible. As the function of ETFβ in BCAA catabolism has been illustrated in Arabidopsis and humans, and N is an essential macronutrient that functions in various stages of plant growth and development[2,4], we speculated that the N nutrition is perhaps the most potent element affecting the variable phenotype of *etfβ*. N is mainly present as nitrate ($NO_3^-$), ammonium ($NH_4^+$), and organic N in soil, and plants prefer inorganic N ($NO_3^-$ and $NH_4^+$) rather than organic N[46]. Accordingly, we achieved the control of the N concentration in the nutrient solution used for soilless cultivation of rice in the greenhouse by replacing the original $NO_3^-$ and $NH_4^+$ ions with chloride and potassium ions, respectively, and the N concentration was adjusted with ammonium nitrate ($NH_4NO_3$) only, while the concentrations of other elements remained normal (1× Hoagland nutrient solution). Solutions containing three different N concentrations were applied to treat *etfβ*: N-free, normal (1 N), and excessive (2 N). Normal nutrient solution contains 0.73 mM $NH_4^+$ and 0.909 mM $NO_3^-$, so we designated the externally applied nutrient solution containing 0.9 mM $NH_4NO_3$ as 1 N and that containing 1.8 mM as 2 N. As expected, we obtained the same results as the previous treatment with three different concentrations (0×, 1×, and 2×) of whole nutrients, further demonstrating that the abnormalities of *etfβ* observed during meiosis are due to N status.

To further explore whether megasporogenesis, like micorsporogenesis in *etfβ*, is affected by N concentration, we also performed similar observations for *etfβ* megasporocytes and embryo sacs. More than 95% megasporocytes of *etfβ* under the 2 N condition [*etfβ* (2 N)] successfully produced normal functional megaspores (Fig. 3a), developing into normal mature embryo sacs; this percentage is close to that observed in WT under the N-free condition [WT (N-free)] (Fig. 3b). In contrast, the percentages observed in *etfβ* under 1 N and N-free conditions [*etfβ* (1 N), *etfβ* (N-free)] were more than 60% and less than 10%, respectively (Fig. 3c). Thus, we conclude that complementation of microsporogenesis and megasporogenesis in *etfβ* is largely dependent on excess N.

**Excessive N supply during the reproductive stage rescues *etfβ* pollen viability.** As meiotic disorders in *etfβ* were rescued by supplying excess N, we measured pollen viability using in vitro and in vivo germination assays. Pollen grains of WT (N-free) and *etfβ* (2 N) germinated well in vitro (Fig. 3d). More importantly, *etfβ* (2 N) pollen germinated normally on its own stigmas and stigmas of WT (N-free). In turn, WT pollens germinated normally on stigmas of both WT (N-free) and *etfβ* (2 N) (Fig. 3e).

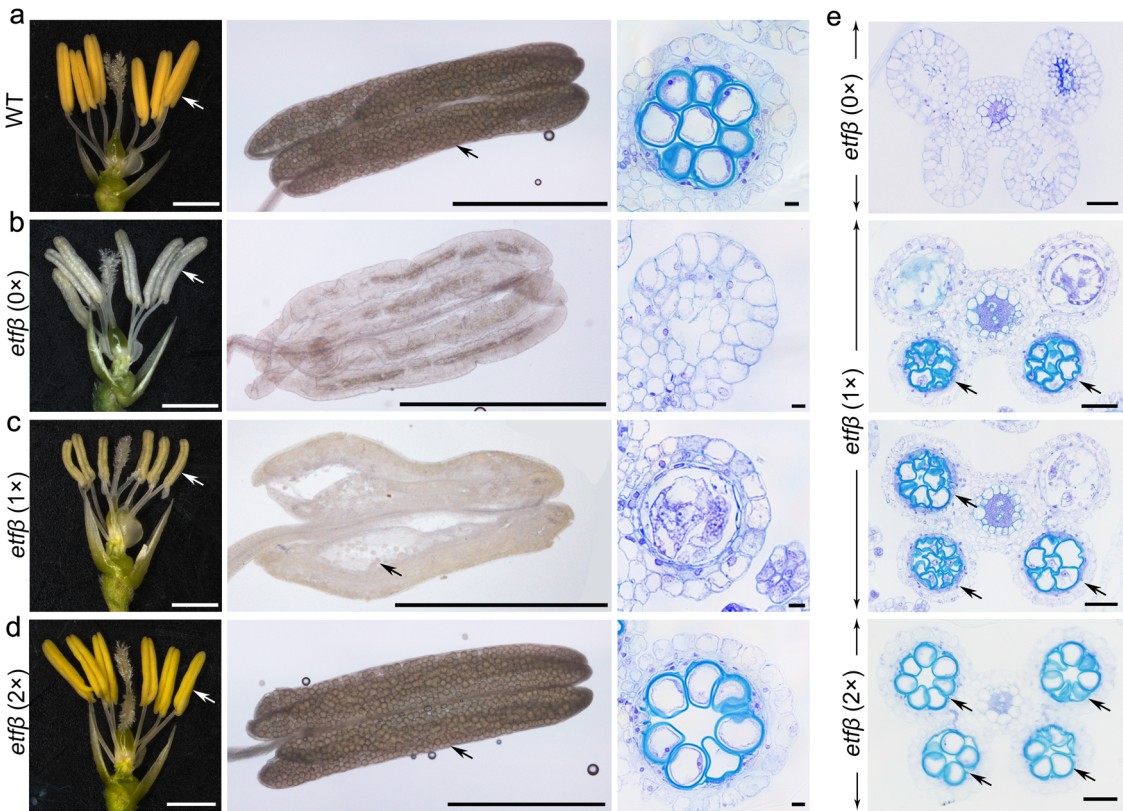

**Fig. 2 Variable phenotype of *etfβ* anthers is affected by nutritional conditions.** Left: spikelet after the removal of lemma and palea. White arrows point to anthers under different nutrient concentrations. Bars = 1 mm. Middle: anther squashes stained with I₂-KI. Black arrows point to pollen grains in anthers. Scale bars, 0.5 mm. Right: transverse sections of anthers at the same developmental stage stained with TBO. Scale bars, 5 μm. **a** Morphology of WT. **b** Morphology of *etfβ* treated with double distilled water [*etfβ* (0×)]. **c** Morphology of *etfβ* treated with normal Hoagland solution [*etfβ* (1×)]. **d** Morphology of *etfβ* treated with twice the normal concentration of Hoagland solution [*etfβ* (2×)]. **e** Phenotypes of the four locules of *etfβ* anthers changed with nutrient concentrations (0×, 1× and 2×). Arrows point to locules within relatively normal pollen grains. Scale bars, 50 μm.

There was no significant difference between the WT and *etfβ* (2 N) pollen in terms of quantity, stigma adhesion, and germination. Thus, excess N not only rescued the defects in *etfβ* meiosis initiation, but also allowed *etfβ* to produce normal pollens.

**Sporogenous cell progenies in *etfβ* fail to acquire meiotic fate under the N-free condition.** To determine the point at which the mutant phenotype is first detectable and to gain detailed insight into the defects of *etfβ* anthers under different N concentrations, we observed and compared anther cells under the two extreme conditions, 2 N and N-free, at different developmental stages. Observation of transverse anther sections stained with TBO and 4′,6-diamidino-phenylindole (DAPI) revealed no significant difference between *etfβ* (2 N) and WT (N-free) from early anther development to the mature pollen stage (Fig. 4a and Supplementary Fig. 6). At the one-layer stage, WT anthers contained only one layer of somatic cells surrounding the central ARs, each containing a large nucleus and nucleolus; in the subsequent two-layer stage, ARs generated sporogenous cells (SCs) and PPCs. After one more round of periclinal division, PPCs generated two layers of secondary parietal cells (SPCs); this stage is referred to as the three-layer stage. The inner layer of SPCs divided periclinally once more to generate the middle layer and the tapetal layer, completing the differentiation of four somatic cells within the anther. The development and differentiation from three- to four-layer stage is crucial for meiosis initiation, the SCs differentiated into PMCs with exceptionally thick cytoplasm, which were much

larger than the surrounding nonreproductive cells and occupied the center of the locules[26]. PMCs progressed through the leptotene, zygotene, pachytene, dyad, tetrad, and pollen stages in sequence.

Abnormalities in anthers of *etfβ* (N-free) were first apparent at the three-layer stage. The central SC-like cells (SCLs) of *etfβ* (N-free) lacked the characteristic thick cytoplasm and large nuclei and nucleoli observed in SCs of WT and *etfβ* (2 N), although the outer three layers of somatic cells were arranged regularly. These SCLs failed to form PMCs, and produced more somatic cells tightly packed in locules (called SC progenies, SCPs). Although the outermost two somatic cell layers could be clearly discerned, the inner two layers were arranged so irregularly that it could hardly be determined whether there were two or three cell layers. In addition, these inner cell layers lacked typical characteristics of either the middle layer or the tapetal layer. Subsequently, the SCPs and inner indeterminate cell layers degenerated and gradually underwent apoptosis. Finally, there was a narrow cavity in every anther locule, leaving only the epidermis. The same results were obtained by observing transverse sections of *etfβ* anthers under N-free and 2 N conditions stained with DAPI (Supplementary Fig. 6).

To further decipher why the SCPs in *etfβ* failed to complete meiosis, we performed RNA in situ hybridization to detect the expression patterns of *MEIOSIS ARRESTED AT LEPTOTENE 1* (*MEL1*) and *RECOMBINATION 8* (*REC8*) in WT and *etfβ* (2 N) anthers. *MEL1* has a specific expression pattern restricted to germ cells, and accordingly serves as a good marker indicating the

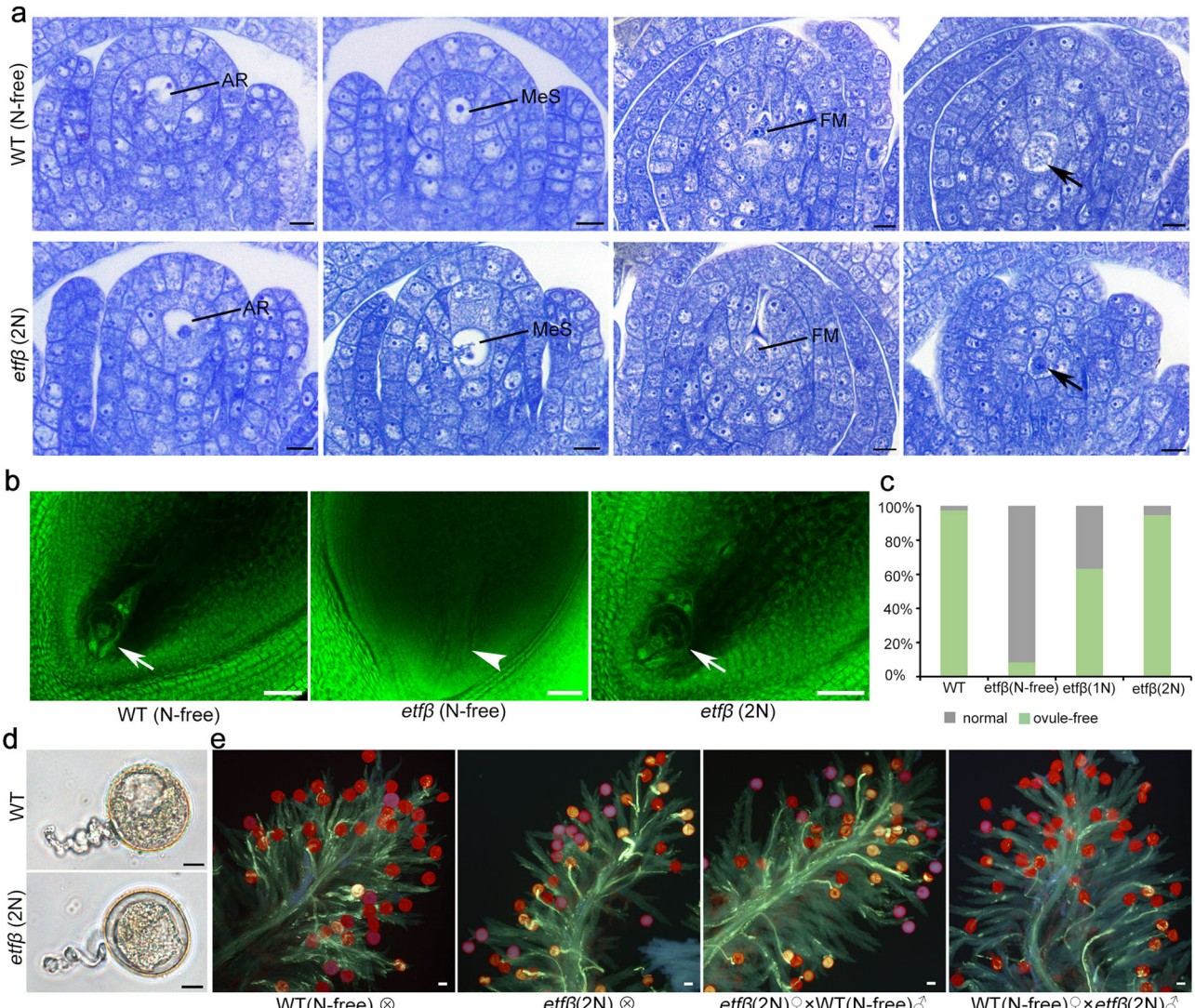

**Fig. 3 Both male and female gametogenesis in *etfβ* can be completely rescued by applying excess N. a** Longitudinal sections of ovules stained with TBO from WT treated with N-free nutritional solution [WT (N-free)] and *etfβ* treated with Hoagland solution with twice the normal N concentration [*etfβ* (2 N)]. AR, archesporial cell; FM, functional megaspore; MeS megasporocyte. Scale bars, 10 μm. **b** Mature ovules of WT and *etfβ* treated with three different concentrations of N: *etfβ* (N-free), *etfβ* (1 N), and *etfβ* (2 N). White arrows point to normal ovules, and the white arrowhead points to an ovule without an embryo sac. Scale bars, 50 μm. **c** The proportion of normal and ovule-free embryo sacs in WT (N-free) and *etfβ* treated with three different concentrations of N: *etfβ* (N-free), *etfβ* (1 N), and *etfβ* (2 N). The *Y*-axis indicates the corresponding ratio. $n \geq 36$. **d, e** Comparison of pollen activity between WT (N-free) and *etfβ* (2 N). **d** In vitro germination of WT and *etfβ* (2 N) pollen. Scale bars, 10 μm. **e** In vivo WT and *etfβ* (2 N) pollen germination and pollen tube elongation. Scale bars, 50 μm. Source data are provided in the Source Data file.

generation of PSCs[26]. In WT (N-free) anthers, a faint *MEL1* mRNA signal was first detected in ARs. Thereafter the signal became stronger in PSCs and SCs, and was persistently strong in the early PMCs (Supplementary Fig. 7 and Fig. 4b). Later, the *MEL1* mRNA signal was undetectable until the onset of meiosis. In *etfβ* (2 N) anthers, the *MEL1* mRNA signal was the same as that in WT (Supplementary Fig. 8 and Fig. 4b). Moreover, the *MEL1* mRNA signal in anthers of *etfβ* (N-free) was the same as that in WT and *etfβ* (2 N) at one-layer stage and two-layer stage (Supplementary Fig. 7). Combined with TBO-and DAPI-stained anther section of *etfβ* (N-free) and *etfβ* (2 N) (Fig. 4 and Supplementary Fig. 6), we speculated there is no obvious developmental defects in ARs and PSC of *etfβ* (N-free). However, the disparity of *MEL1* mRNA signal in *etfβ* (N-free) compared with *etfβ* (2 N) and WT was first detected in SCLs at three-layer

stage. The *MEL1* mRNA signal in SCLs of *etfβ* (N-free) anthers was so weak and faded so quickly that it was almost undetectable in SCPs until the four-layer stage (Fig. 4b).

*REC8*, which encodes a meiosis-specific cohesion element[26], is preferentially expressed in SCs and microsporocytes of WT and *etfβ* (2 N) (Supplementary Fig. 8 and Fig. 4c). However, no *REC8* signal above background was detected in SCLs and SCPs of *etfβ* (N-free) (Fig. 4c), indicating that the SCPs did not differentiate into microsporocytes. We therefore hypothesized that SCPs in *etfβ* fail to acquire meiotic fate under N-free conditions, and the abnormal expression patterns of *MEL1* and *REC8* in SCLs of *etfβ* (N-free) were responsible for the failure of meiosis initiation and the generation of SCPs without meiotic fate. Moreover, the meiotic initiation defects in *etfβ* can be rescued by excess exogenous inorganic N, and ETFβ is specifically involved in meiotic initiation.

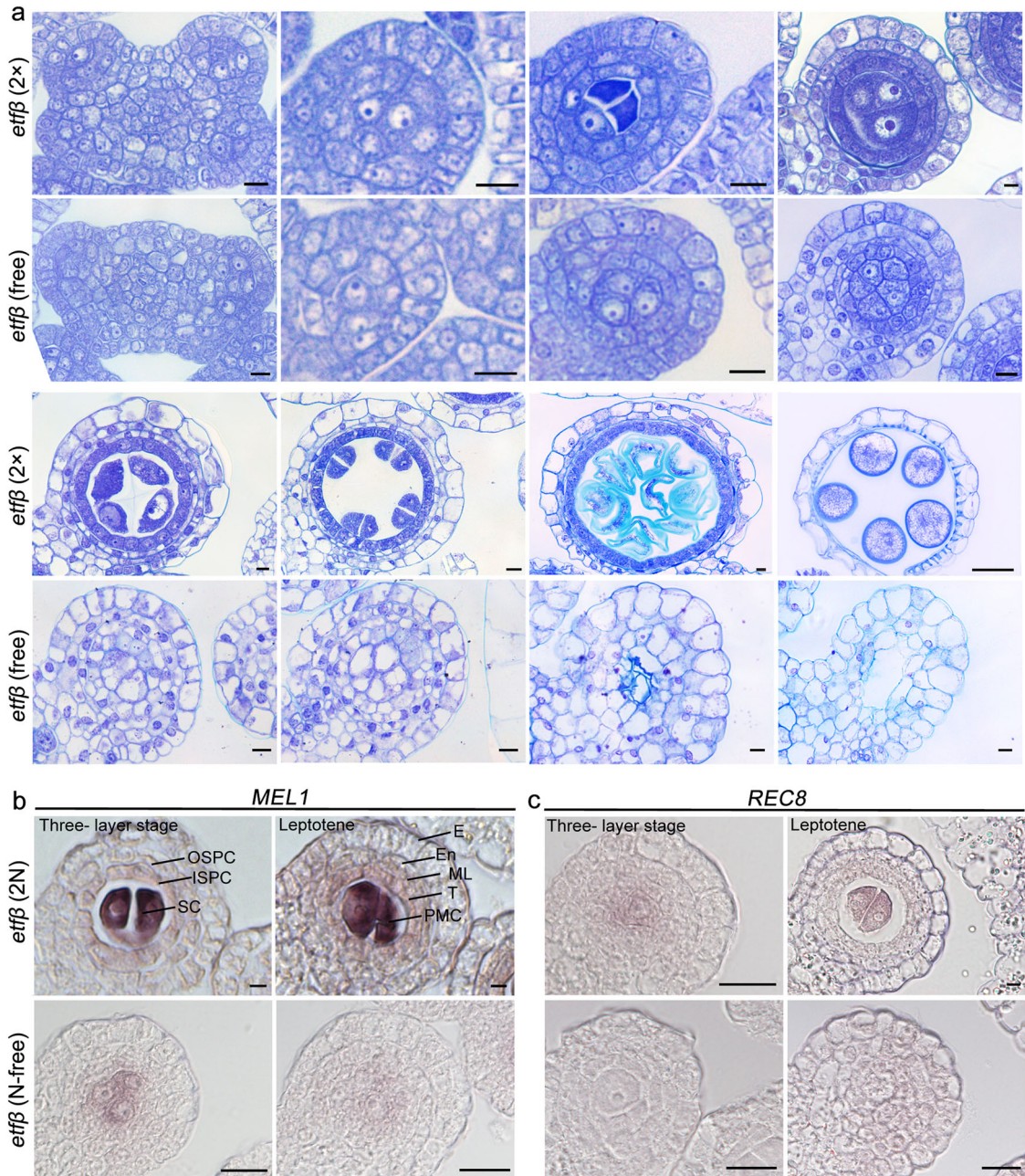

**Fig. 4 Sporogenous cell progenies in *etfβ* fail to acquire meiotic fate under N-free conditions. a** TBO-stained transverse sections of anthers of *etfβ* (2 N) and *etfβ* (N-free). The arrows indicate abnormal apoptosis in the center of locules. SCL sporogenous cell-like cell, SCP sporogenous cell progeny. Scale bars, 5 μm. **b** In situ expression analysis of *MEL1* in *etfβ* (2 N) and *etfβ* (N-free) anthers. Scale bars, 5 μm. **c** In situ expression analysis of *REC8* in *etfβ* (2 N) and *etfβ* (N-free) anthers. Scale bars, 5 μm.

**ETFβ is involved in BCAA catabolism, and the abnormal metabolite levels in *etfβ* can be rescued by applying excess exogenous N.** The Arabidopsis ETFβ is characterized to be involved in BCAA catabolism[29]. Therefore, we investigated the relationship between reproductive growth and metabolism in *etfβ* and WT under three N concentrations (N-free, 1, and 2 N). The primary metabolite contents in 9-cm (±1 cm) and 18-cm (±1 cm) panicles of *etfβ* and WT under the three N concentrations were measured. As the N concentration decreased, a large amount of amino acids and their derivatives accumulated significantly (Fig. 5). The metabolic abnormalities (either an increase or decrease in content) worsened as the growth stage advanced (from 9 to 18 cm). This worsening of metabolic abnormalities in *etfβ* (N-free) as panicles progressed from 9 to 18 cm is reasonable,

as the cytological phenotypes of *etfβ* (N-free) were aggravated from early meiosis to later stages (Fig. 4a).

Among the metabolites, Leu and Ile (marked with red letters and asterisks in Fig. 5) are BCAAs; the contents of these amino acids in *etfβ* (N-free) were significantly higher than those in WT (N-free). The sharp accumulation subsided with the increasing N concentration, as the pink curves approached the black curves as the X-axis extends (Fig. 6a, b). The contents of threonine (Thr), lysine (Lys), aspartate (Asp), and methionine (Met) also changed (Fig. 6c-f). Lys, Met, Thr, and Ile are produced from a common precursor, namely Asp, via a branched and complicated pathway, and are commonly known as Asp-family amino acids[47,48]. Thr accumulated abnormally in *etfβ* (N-free), as a result of the accumulation of Ile via feedback inhibition of Thr deaminase[49].

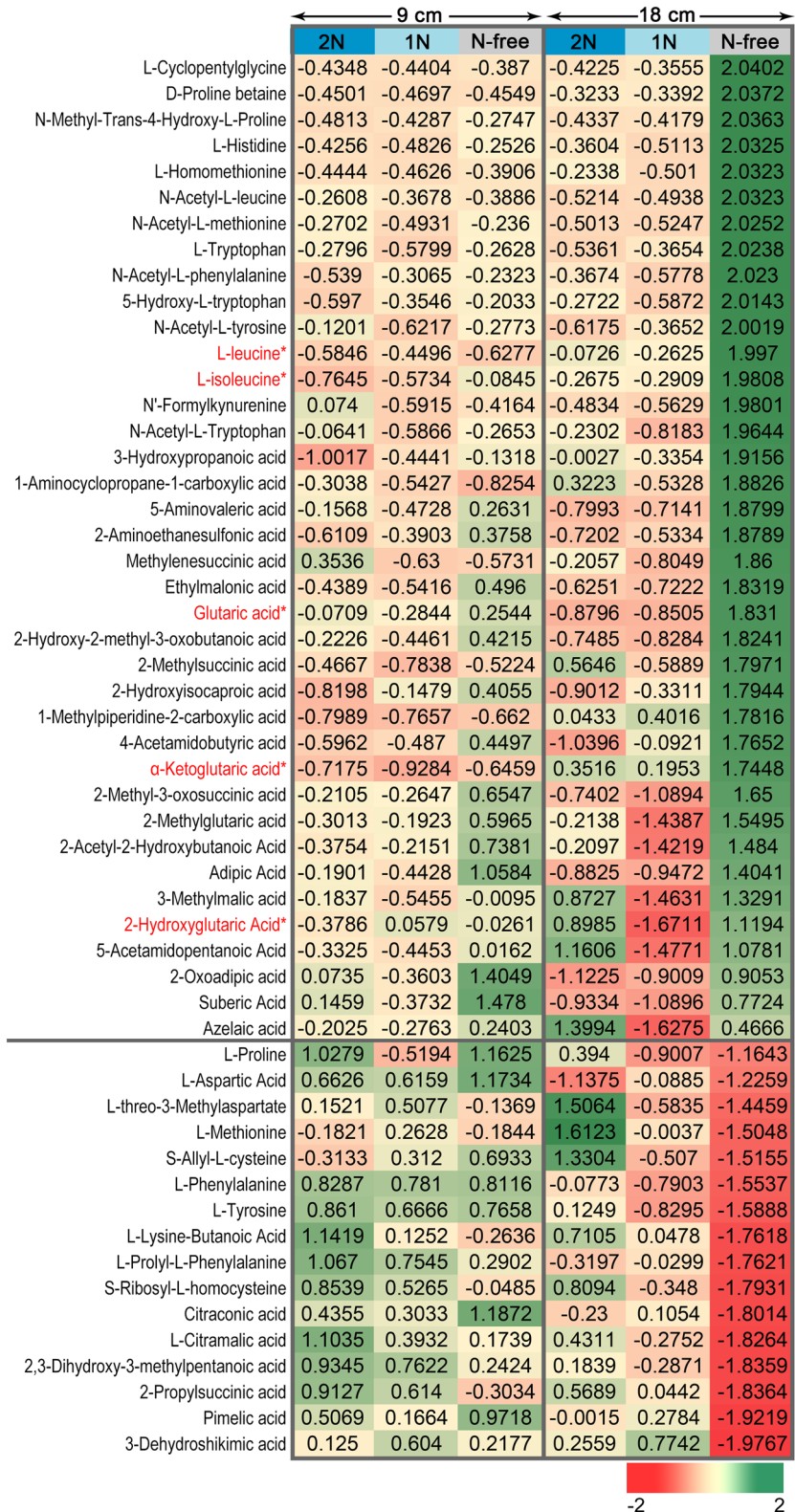

**Fig. 5 Alteration in the contents of amino acids and derivatives and organic acids in *etfβ* relative to WT is exacerbated by a decrease of N supply.** Fold changes of metabolite content in *etfβ* relative to WT are standardized to (−2, 2) by *z*-score. Asterisks at the upper right of the metabolite names marked in red indicate known metabolic substrates of the ETF/ETFQO system. Metabolites above the black line are significantly higher in *etfβ*, and metabolites below the black line are markedly lower in *etfβ*. The color bar on the top of the heatmap indicates N concentration: bright blue, 2 N; wathet blue, 1 N; and gray, N-free. Each color block represents the fold change of the corresponding metabolite content in *etfβ* relative to WT under the same N concentration at the same stage (panicle length of 9 cm, the left three columns; panicle length of 18 cm, the right three columns). Three independent biological repeats were performed. Source data are provided in the Source Data file.

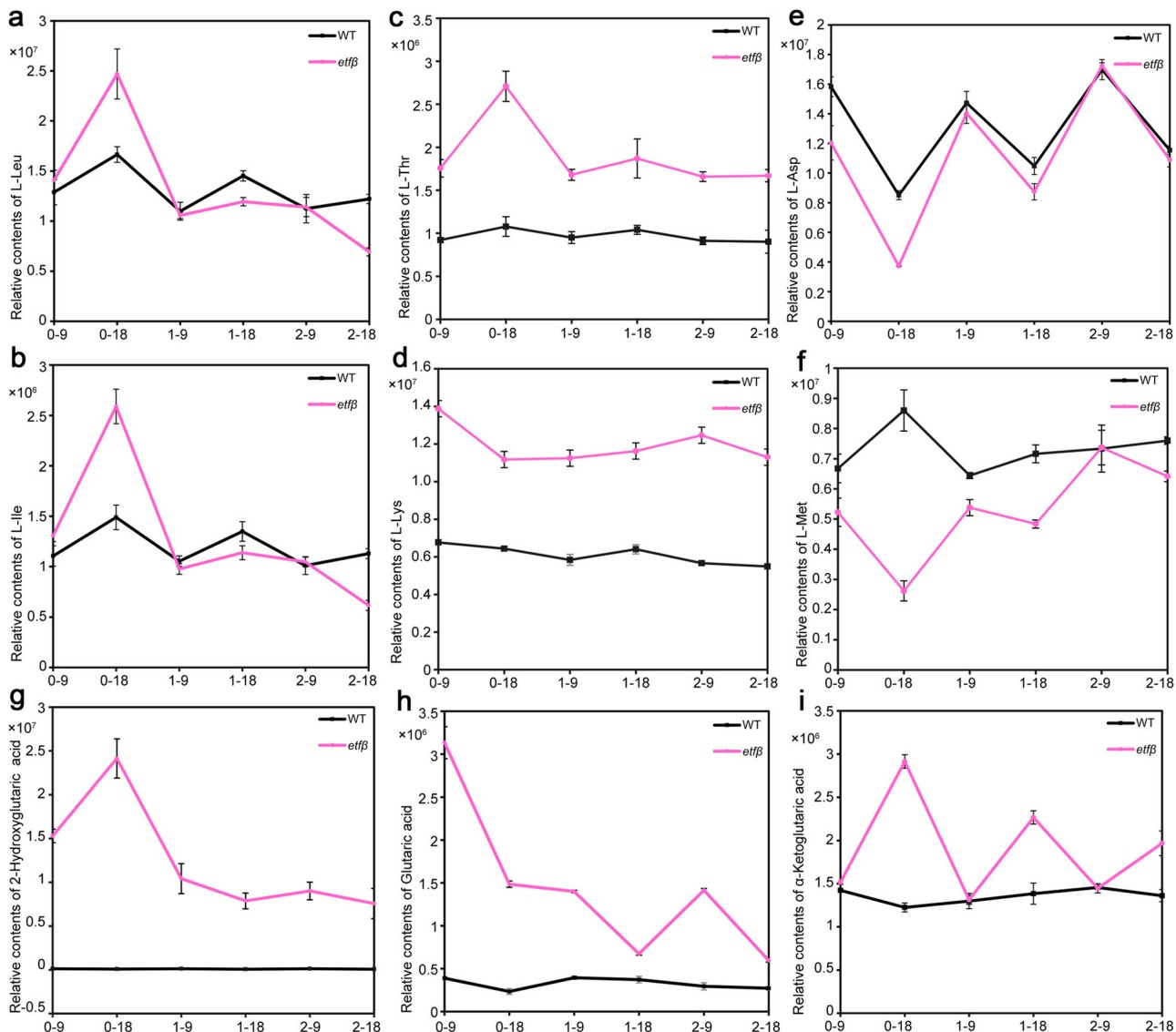

**Fig. 6 Nitrogen deficiency resulting from metabolic abnormalities in *etfβ* can be rescued by increasing exogenous N. a–f** Relative contents of L-Leu, L-Ile, L-Thr, L-Lys, L-Asp, and L-Met in wild type (WT) and *etfβ* under three N concentrations and two developmental stages. The numbers on the *X*-axis, such as 0–9, 0–18, 1–9, 1–18, 2–9 and 2–18, indicate the corresponding N concentrations (0: N free, 1: 1 N, and 2: 2 N) and developmental stages (9 and 18 cm panicles in length). Error bars indicate standard deviation of three independent biological replicates. **g–i** Relative contents of glutaric acid, 2-hydroxyglutaric acid, and α-ketoglutaric acid in WT and *etfβ* under three N concentrations and two developmental stages. Error bars indicate standard deviation (SD) of three independent biological replicates. Source data are provided in the Source Data file.

The accumulation of Thr in *etfβ* (N-free) led to a rapid decrease in Met because of the negative feedback inhibition of homoserine dehydrogenase (HSDH). There are three competing pathways controlled by HSDH: the Thr, Met, and Lys biosynthesis pathways[50]. As the fluxes toward the Thr and Met biosynthesis pathways weakened, the flux towards Lys biosynthesis was accordingly enhanced and the contents of Lys in *etfβ* (N-free) went up. Thus, there is a reasonable explanation for the changes in the contents of these amino acids in *etfβ* as the N concentration changes.

In addition, the concentrations of three organic acids, 2-hydroxyglutaric acid, glutaric acid, and α-ketoglutaric acid, also changed with a change in N concentration (Fig. 6g–i). 2-hydroxyglutaric acid is a known substrate of D2HGDH[43], and glutaric acid and α-ketoglutaric acid are the derivatives of 2-hydroxyglutaric acid. The contents of these organic acids in WT remained at very low levels regardless of the N

concentration and developmental stage. Interestingly, a sharp increase in the accumulation of these organic acids occurred in *etfβ* (N-free) (marked with red letters and asterisks in Fig. 5), indicating a severe disruption of D2HGDH caused by the loss of ETFβ.

To unravel the further molecular function of ETFβ and ETF/ETFQO system in rice BCAA catabolism, we performed real-time PCR analysis to investigate the expression profiles of all five genes encoding components of the ETF/ETFQO system in *etfβ* and WT growing under the same N concentrations (Fig. 7a). The *ETFβ* transcript was undetectable in *etfβ*, and the expression levels of the other four genes, *ELECTRON TRANSFER FLAVOPROTEIN SUBUNIT α* (*ETFα*), *ETFQO*, *IVDH*, and *D2HGDH* and the two meiosis marker genes *MEL1* and *REC8* decreased significantly compared with WT, suggesting that functional defects of ETFβ impair the function of the ETF/ETFQO metabolic pathway. The results of real-time PCR analysis are consistent with the significant accumulation of

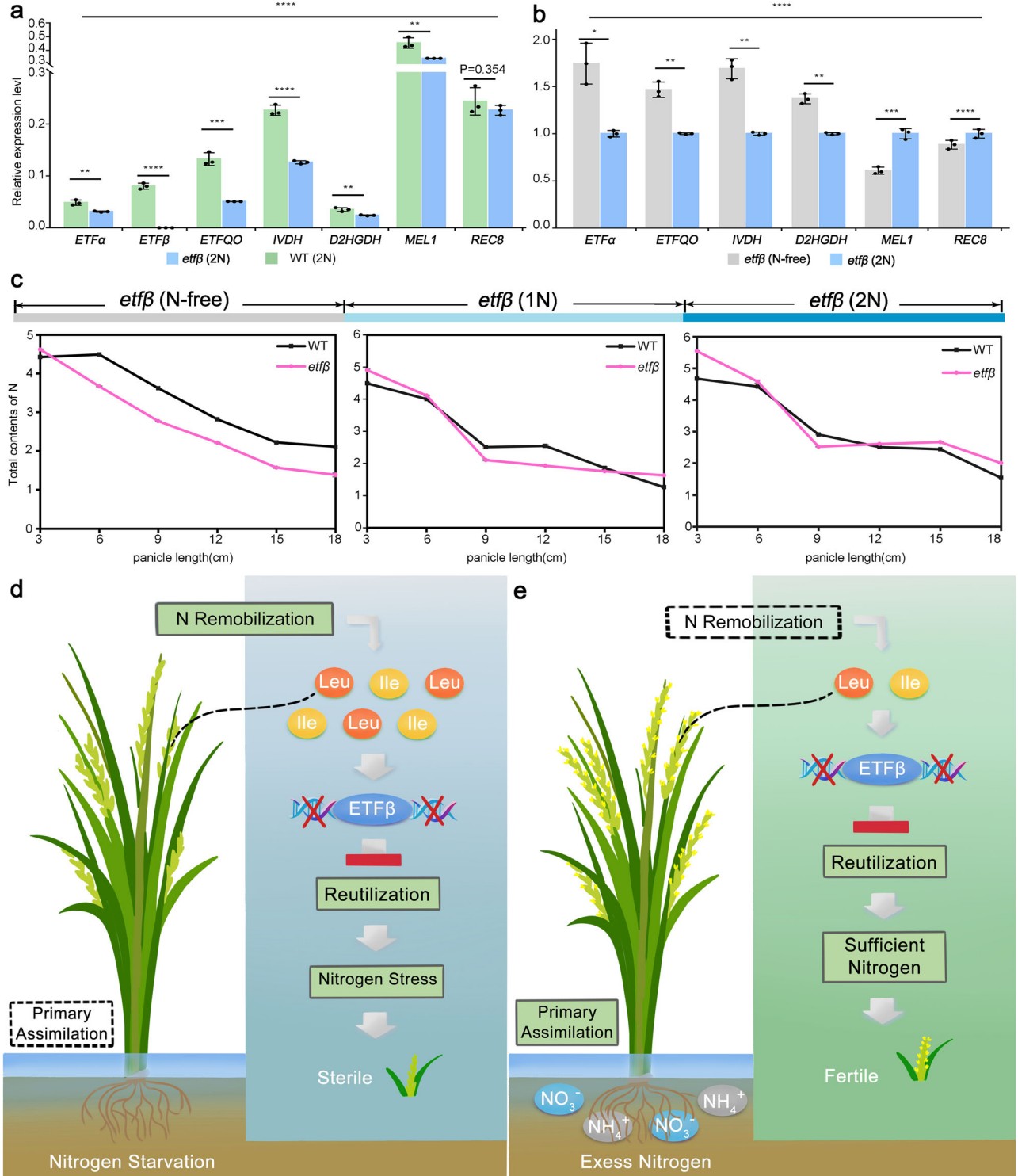

**Fig. 7 N nutrition contributes to plants fertility by determining meiosis initiation. a** Relative expression of *ETFα, ETFβ, ETFQO, IVDH, D2HGDH,* and two meiosis marker genes, *MEL1* and *REC8*, in WT (2 N) and *etfβ* (2 N) anthers. **b** Relative expression of *ETFα, ETFQO, IVDH, D2HGDH, MEL1*, and *REC8* in *etfβ* (2 N) and *etfβ* (N-free) anthers. Error bars indicate SD of three independent biological replicates in real-time PCR (**a**, **b**). Significance was determined by two-tailed Student's *t*-test. ***p* < 0.01, ****p* < 0.001; *****p* < 0.0001 (**a**, b). **c** Total N contents of panicles in WT and *etfβ* at six developmental stages (3, 6, 9, 12, 15, and 18 cm panicles in length) were measured under three N concentrations (N-free, 1, 2 N). The color bar indicates N concentration. Error bars indicate SD of three independent biological replicates. **d, e** Proposed model of ETFβ function. **d** Sterility of *etfβ* results from N deficiency in developing panicles under low N-stress conditions. **e** Meiosis initiation defects in *etfβ* could be rescued by supplying excess exogenous inorganic N. Source data are provided in the Source Data file.

2-hydroxyglutaric acid and its derivatives (Fig. 6g–i) and the known function of *Arabidopsis* D2HGDH[43]. We conclude that rice ETFβ is involved in the catabolism of BCAAs (mainly Leu and Ile) and organic acids, and applying exogenous N. can rescue that abnormal metabolite levels in *etfβ*.

The ETF/ETFQO system has been reported to operate in plants under stress conditions and provides an alternate electron supply to the mitochondrial electron transport chain to succinate;[29] thus, it would be interesting to investigate whether this alternate pathway is upregulated under N stress conditions. The real-time PCR analysis of *etfβ* under 2 N and N-free conditions suggested that transcription of five genes of the ETF/ETFQO system can be induced by low N stress (Fig. 7b). We speculate that under N-stress conditions, the ETF/ETFQO system plays even more important roles, or that plants are more dependent on this metabolic pathway to catabolize BCAAs as carbohydrate substrates for respiration are limiting. As an alternative metabolic pathway, the ETF/ETFQO system increases the metabolic flexibility during reproduction for plants under N stress conditions.

**N deficiency resulting from metabolic abnormalities causes the meiotic initiation defects in *etfβ*, which can be rescued by supplying excess N.** Loss of function of ETFβ leads to not only accumulation of BCAAs, but also decrease in other amino acids to different degrees under N-free conditions. To further evaluate the contribution of ETFβ to N remobilization and reutilization in rice during reproduction, the total N contents of panicles at six different stages (panicle lengths of 3, 6, 9, 12, 15, and 18 cm) were measured under the three N concentrations (Fig. 7c). As expected, the total N contents of panicles in *etfβ* (N-free) were always significantly lower than those of WT. However, the differences between *etfβ* mutant and WT under 1 N and 2 N conditions, especially 2 N, were not as remarkable as those under the N-free condition. N deficiency in *etfβ* panicles could also be rescued by increasing exogenous inorganic N, as the meiosis initiation defects and abnormal metabolic levels do.

These findings suggest that the ETF/ETFQO system contributes to N remobilization and reutilization, thereby increasing NUE, allowing the limited N to exert greater effects and ensuring meiosis initiation and maintenance of fertility.

## Discussion

ETFβ and the alternative metabolic pathway, the ETF/ETFQO system, have been detected in various organisms as early as in the last century[34–37]. GA II in humans[39,40] and abnormal accumulation of BCAAs in Arabidopsis[41] emphasize the crucial role of ETFβ in BCAA catabolism. Here, we demonstrate that ETFβ can support meiotic initiation by contributing to N reutilization, showing an essential function for this protein in plant sexual reproduction.

We propose a functional model for ETFβ under different soil conditions (Fig. 7d, e). When exogenous inorganic N is sufficient, primary assimilation and reutilization can provide a double guarantee that the demands of N sink tissues will be met[51]. Yet, the primary assimilation pathway is blocked under N-deficient conditions, and only through N reutilization can plants meet the growth needs of N sink tissues, and thus ensure sexual reproduction[52]. Nitrogenous substances, including proteins in mature N sources, are hydrolyzed to amino acids, and further metabolized through the alternate pathway, ETF/ETQO system; N can thus be remobilized and reutilized, explaining how rice plants maintain high fertility under N starvation. Leu and Ile can be catabolized through the ETF/ETFQO system. Based on the above results, we conclude that the loss of function of ETFβ leads to the accumulation of Leu and Ile and their derivatives; meanwhile, N is fixed in these amino acids and cannot be remobilized

and reutilized. N reutilization is impaired, exacerbated by exogenous N deficiency, consequently the N demands of sink tissues cannot be met at the appropriate time. As the main N sink during reproduction, spikelets show a sterile phenotype because of N deficiency under low N stress (Fig. 7d). However, plants can meet the demands of these N sink tissues via primary assimilation instead of N reutilization when exogenous organic N is sufficient; there is no need to catabolize large amounts of protein for N reuse[52]. Therefore, even if BCAAs cannot be normally catabolized due to functional defects in the ETF/ETFQO system, they will not accumulate in large amounts. Consequently, meiosis initiation, fertility, metabolite levels, and total N contents in *etfβ* can be restored under 2 N conditions (Fig. 7e). The ETF/ETFQO system increases the flexibility of this vital metabolic process, and contributes to ensuring meiotic initiation by promoting N reutilization under N starvation, which can be a safeguard mechanism for plants to maintain fertility in poor soil. Therefore, we conclude that sufficient N nutrition from either reutilization or primary assimilation is a prerequisite for rice meiosis initiation.

It is worth mentioning that the dependence of rice meiosis initiation on N nutrition is different from the crucial effects of nutritional stress in budding yeast[23] and RA in mice[24]. Yeast meiosis produces spores that may survive harsh nutritional conditions. The dependence of yeast cell division on nutritional status can be considered a survival strategy allowing for the optimal usage of available nutrients[22]. In mice, RA signaling determines germ cell fate, accelerating the entry of germ cells into meiotic prophase[21]. Nutritional stress in yeast and RA signaling in mice for meiosis initiation are synergistically regulated by nutritional cues and the genome[18], but not explicitly and directly related to nutrient reuse in vivo. However, in *etfβ* under low N stress conditions, deficiency of exogenous inorganic N synergizes with the functional defect in *ETFβ* to create a special N stress condition whereby both primary assimilation and the reutilization of N are severely weakened. Under the special N stress condition, *etfβ* exhibited meiosis initiation defects, which cannot be observed in normal rice plants under low N stress conditions or *etfβ* under N excess conditions.

So far, various environmental factors have been shown to affect crop fertility and yield[53–55]. The dynamic molecular networks in plants employed to ensure development in response to environmental changes have attracted more and more attention. We demonstrate that soil N nutrition is a key environmental factor that maintains plant fertility by affecting meiosis initiation. Our finding promotes further understanding of the relationship between N nutrition and meiosis initiation, and provides insights into how plants overcome barren soil conditions to achieve normal fertility via genotype–environment interactions, revealing new possibilities for engineering plants to maintain fertility stability in soil environments without sufficient nutrition.

## Methods

**Plant materials and growth conditions in paddy field.** The *etfβ* mutant was identified from an indica rice cultivar Zhongxian 3037 treated by [60]Co γ-ray radiation, and the mapping population was derived from the cross of heterozygous mutant plants with Zhonghua 11 (a japonica rice variety). And Zhongxian 3037 was used as WT. Plants were grown under natural and normal conditions in experimental paddy fields in Changping (40.2°N, 116.2°E), Beijing during summer and Lingshui (18.5°N, 110.0°E), Hainan province during winter, respectively.

**Map-based cloning of *ETFβ* and complementation in *etfβ*.** The *etfβ* was isolated from Zhongxian 3037 (an indica rice variety) and induced by [60]Co γ-ray irradiation. The mapping population was derived from the cross of heterozygous *etfβ* plants with Zhonghua 11. Leaves from both parents and sterile plants of the F2, F3 and F4 population were collected to extract DNA. The InDel markers were developed based on differences in DNA sequences of the total genome between the two parents. Primers were designed using PRIMER 6 software. Primer sequences of the InDel markers used for map-based cloning and genotyping assays are given in

Supplementary Table 1. F2 and F3 progenies were used for linkage analysis and fine mapping. 97 F2 plants for first-pass mapping. M1 and M2 (Supplementary Table 1) located on the short arm of chromosome 4 were linked with the target gene and mapped onto a 2.33-Mb region. 214 F3 and 507 F4 population refined the linkage region to 170 kb via six InDel markers (Supplementary Table 1, Supplementary Data 1) including 14 genes. Sequence comparisons between the mutant plants and WT were then performed for all the annotated genes within this region. A large fragment deletion of 1922 bp was sequenced (Supplementary Fig. 2) including the 5′-untranslated region (UTR), the first and second exons in LOC_Os04g10400, and ETFβ transcript was almost undetectable in the mutant (Supplementary Fig. 1c, d).

We created the complementation construct pCMIL by inserting a 5.5-kb genomic DNA fragment containing the entire ORF of ETFβ, 5.0-kb upstream region and 1.8-kb downstream region to the binary vector pCAMBIA1300 (CAMBIA). And the plasmid was introduced into calli derived from the seeds of heterozygous plants for the ETFβ locus using an Agrobacterium tumefaciens-mediated method[56]. The construct was introduced to the Agrobacterium strain EHA105, and transformed into calli from seeds of Zhongxian 3037 to generate the desired complementary line of etfβ. The hygromycin-resistant regenerators were transplanted into the soil and grown either in a greenhouse or in the field.

**Phenotypic characterization**. Histological studies of anther and ovule development were performed by conventional assays[26]. Fresh young panicles of both WT and etfβ were fixed in Carnoy's solution (ethanol:glacial acetic, 3:1), spikelets of WT and etfβ were dehydrated using graded ethanol and embedded in Technovit 7100 resin (Hereaus Kulzer, Wehrheim, Germany). Four-micrometer transverse and longitudinal sections were acquired using a Leica microtome, stained with 0.25% toluidine blue O (TBO, Sigma-Aldrich, Aldrich, St. Louis, MO, USA) or 4′,6-diamidino-phenylindole (DAPI, Vector Laboratories, Burlingame, CA, USA). The anthers were dyed using the $I_2$-KI solution. Anthers sections stained with TBO and anthers stained with $I_2$-KI was observed under microscopy Olympus BX 51. Anthers stained with DAPI was observed with a Zeiss A2 fluorescence microscope with a micro-CCD camera.

Mature ovules of WT and mutant plants were fixed for 10 h in FAA stationary liquid (ethanol:water:glacial acetic acid:formaldehyde = 4.75:3.75:0.5:1), followed by washes of 70%, 50%, and 30% ethanol and a final rinse with distilled water. Fixed ovules were treated with a 2% aluminum potassium sulfate solution for 20 min and stained in a solution of eosin B dissolved in 4% sucrose at a concentration of 10 mg/L for 10 h. The ovules were dehydrated with ethanol. Subsequently, ovules were transferred into a mixture of ethanol and methyl salicylate (1:1) for 1 h and cleared in pure methyl salicylate overnight. Images were acquired with a confocal laser-scanning microscope at an excitation wavelength of 405 nm (Leica TCS SP5, Wetzlar, Germany).

**Subcellular localization**. The ETFβ CDS sequences were obtained by PCR using Zhongxian 3037 cDNA. The ETFβ CDS sequence was cloned into a pJIT163-GFP vector[9] to generate an ETFβ–GFP fusion construct, and the ETFβ CDS sequence was cloned into a pJIT163-mCherry vector (obtained by replacing the GFP of pJIT163-GFP with mCherry) to generate an ETFβ-mCherry construct. The constructs were transformed into rice protoplasts. Transformed protoplasts were incubated in the dark at 28 °C for 20 h. Fluorescence signals were captured as images with a confocal laser-scanning microscope (Leica TCS SP5, Wetzlar, Germany). Primers are listed in Supplementary Table 2.

**In situ hybridization**. Fresh panicles, which were still embedded in the flag leaves, were dissected from plants, and were fixed in formaldehyde–acetic acid–ethanol solution (50% v/v ethanol, 5% v/v acetic acid, 3.7% v/v formaldehyde) for 16 h at 4 °C, and dehydrated by passing through an ethanol series (50–60–75–80–95–100–100%). The samples were then passed through a xylene–ethanol series, then embedded in paraplast (Sigma-Aldrich, St. Louis, MO, USA) with six changes of paraplast. The blocks were sectioned into 8 μm slices using an RM 2235 (Leica, Wetzlar, Germany) rotary microtome and mounted onto Poly-Prep slides (Sigma-Aldrich, St. Louis, MO, USA). The sections were hybridized to either a sense or an antisense probe of ETFβ, MEL1, and REC8. Probes were synthesized via PCR. A 188 bp ETFβ cDNA fragment (nucleotides 1–188, counted from the transcription start codon) was amplified from panicle-derived cDNA using primers incorporating the T7 polymerase binding site. The cDNA fragment of MEL1 probe was 213 bp (nucleotides 3002–3124, counted from the transcription start codon). The cDNA fragment of REC8 probe was 235 bp (nucleotides 1–235, counted from the transcription start codon). Primer sequences are listed in Supplementary Table 2. Gene-specific segments were amplified and cloned into pEASY-Blunt simple vector (Transgen Biotechnology, Beijing, China)[26]. The sense and antisense probes were synthesized with T7 RNA polymerase (Roche Diagnostics, Mannheim, Germany using the digoxigenin RNA labeling kit (Cat. 11175041910, Roche Diagnostics, Mannheim, Germany), following the manufacturer's instructions. The sections were deparaffinized in xylene, rehydrated through a graded ethanol series and then air-dried. They were treated with 4 μg mL⁻¹ proteinase K in 100 mM Tris–HCl, pH 8.0, 50 mM EDTA at 37 °C for 30 min and then washed twice with distilled water for 5 min each. The sections were washed as following: twice with 1 × PBS (130 mM NaCl, 7 mM $Na_2HPO_4 \cdot 7H_2O$, 3 mM $NaH_2PO_4 \cdot H_2O$) for 2 min, 0.2% glycine in 1 × PBS for 2 min, twice with 1 × PBS for 2 min. They were subsequently treated with 40 mL 1 × PBS with 0.5% acetic anhydride and 100 mM triethanolamine for 5 min at room temperature, washed twice in 1 × PBS for 5 min. Sections were

incubated at 50 °C for 14–16 h with coverslips in hybridization buffer (100–120 μl per slide) containing the probes (0.02 μg mL⁻¹). The hybridization buffer consisted of 50% deionized formamide, 0.3 M NaCl, 10% dextran sulfate, 10 mM Tris–HCl, pH 7.5, 1 mM EDTA, 100 mM DTT and 500 μg mL⁻¹ Escherichia coli tRNA. After hybridization, successive washing steps were performed as follows: twice in 0.2 × SSC, once in RNase A (5 μg mL⁻¹) in STE (0.5 M NaCl, 10 mM Tris–HCl, 5 mM EDTA, pH 7.4) at 37 °C for 30 min, twice in STE at 37 °C for 5 min each, and twice in 0.2 × SSC at 55 °C for 30 min each. Immunological detection of the hybridized probes was performed according to the manufacturer's manual with some modifications. The slides were soaked with 1 × TBS (150 mM NaCl, 100 mM Tris–HCl, pH 7.5) and then incubated with 1 × blocking reagent (Roche) in TBS for 45 min. They were further incubated with TBST (1% BSA and 0.3% Triton X-100 in TBS) for 45 min, followed by incubation with the diluted antidigoxigenin alkaline phosphatase conjugate (1:1250) in TBST for 2 h. The slides were subsequently washed four times with TBST for 15 min. The sections were rinsed with reaction buffer (100 mM NaCl, 50 mM $MgCl_2$, 100 mM Tris–HCl, pH 9.5), and then covered with reaction buffer contains 0.34 mg mL⁻¹ nitro blue tetrazolium salt and 0.175 mg mL⁻¹ 5-bromo-4-chloro-3-indolyl phosphate toluidinium salt. After incubation at room temperature for 12 h in the dark, the color reaction was stopped by immersing the slides in TE (10 mM Tris–HCl, 1 mM EDTA, pH 7.5). The images were acquired by microscopy Olympus BX 51.

**Hydroponic culture conditions and gradients setting**. Hydroponic culture was performed in a phytotron at a temperature of 23–35 °C with 10 h light and 14 h dark photoperiod, about 70% humidity and about 200 μmol m⁻² s⁻¹ photon density. Plants were planted in porous ceramic (Profile) cultivated with Kimura B nutrient solution (usually as normal N condition) including macronutrients (0.37 mM $(NH_4)_2SO_4$, 0.18 mM $KNO_3$, 0.37 mM $Ca(NO_3)_2$, 0.18 mM $KH_2PO_4$, 0.09 mM $K_2SO_4$, 0.55 mM $MgSO_4 \cdot 7H_2O$, 1.6 mM $Na_2SiO_3 \cdot 9H_2O$) and micronutrients (46.2 μM $H_3BO_3$, 0.32 μM $CuSO_4 \cdot 5H_2O$, 9.14 μM $MnCl_2 \cdot 4H_2O$, 0.08 μM $(NH_4)6Mo_7O_{24} \cdot 4H_2O$, 0.76 μM $ZnSO_4 \cdot 7H_2O$, and 40 μM Fe(II)-EDTA), pH 5.75[5]. And the nutrient solution was replaced every three days. Porous ceramics were purchased from PROFILE Products LLC.

For gradients treatment with different N concentrations, $(NH_4)_2SO_4$ and $KNO_3$ were replaced by $NH_4NO_3$, and $Ca(NO_3)_2$ was replaced by $CaCl_2 \cdot 2H_2O$. We used 0, 0.9, and 1.8 mM $NH_4NO_3$ for hydroponic culture. In comparison with the N concentration of Kimura B solution (1.7 mM), 0 mM $NH_4NO_3$ was used as N free condition, 0.9 mM $NH_4NO_3$ was used as moderate N, and 1.8 mM $NH_4NO_3$ was used as excessive N in this study.

**In vitro pollen germination and pollen tube growth**. Fresh pollen grains were collected from at least three WT and etfβ plants grown under three different N gradients. Then pollen grains were transferred into a liquid germination medium [20% (w/v) sucrose, 10% (v/v) polyethylene glycol 4000, 3 mM $Ca(NO_3)_2 \cdot 4H_2O$, 40 mg L⁻¹ $H_3BO_3$, and 3 mg L⁻¹ vitamin B1] and cultured for about 10–20 min at room temperature (30 °C) under moist conditions to generate synchronously germinated rice pollen grains. At least three experiments were conducted. The observations of reproductive organs and the activity of pollen grains were performed as described[57]. Images were acquired under microscopy Olympus BX 51.

**Aniline blue staining of pollen tubes in vivo**. Decolorized Aniline Blue solution was prepared by making a 0.1% (w/v) solution of Aniline Blue (Acros Organics), adding 1 M NaOH drop-wise, and incubating the solution overnight at 48 °C until it became a transparent yellow color. Following in vivo pollination, mature flowers in wild-type and etfβ were pollinated with wild-type, etfβ pollen, respectively. After 10, 20, and 30 min, the pollinated pistils were removed from the plants and incubated in fixing solution containing ethanol:acetic acid (3:1) for 2 h at room temperature. The fixed pistils were then washed with distilled water three times for 5 min each followed by incubation in 1 M NaOH softening solution overnight. Pistils were placed into decolorized Aniline Blue solution and allowed to stain for 12–24 h before being mounted on slides and observed using 405/440–480-nm setting or fluorescent microscopy[57]. Images were acquired using a Zeiss 780 two-photon laser scanning confocal microscope (CLSM; Carl Zeiss).

**Quantitative real-time RT-PCR**. Total RNA was isolated from 6 cm (±1 cm) panicles in WT and etfβ grown under three different N gradients. After snap-freezing the material in liquid N, RNA isolation and cDNA synthesis was carried out via standard methods using TRIZOL reagent and SuperScript® III Reverse Transcriptase (Invitrogen). Real-time PCR analysis was performed on a BioRad CFX96 using Subgreen (Biotium, CA, USA) using gene-specific primers for quantifying the ETF/ETFQO system and meiosis initiation genes expression, respectively (http://rice.plantbiology.msu.edu/index.shtml). OsACTIN was used as a normalizer (for normalization corresponding to the total RNA level) in the RT-qPCR assays. Primers used for real-time PCR are listed in Supplementary Table 2.

**Metabolomics analysis**. For metabolomics analysis, samples of two gradients (9 and 18 cm long panicles ± 1 cm) from six independent WT and etfβ under three different N treatments were sampled and ground in liquid nitrogen. Sample preparation and metabolomics analysis were completed by Metware, Wuhan, China (https://www.metware.cn/)[58].

Biological samples were freeze-dried by vacuum freeze-dryer (Scientz-100F). The freeze-dried samples were crushed using a mixer mill (MM 400, Retsch) with a zirconia bead for 1.5 min at 30 Hz. Dissolved 100 mg of lyophilized powder with 1.2 ml 70% methanol solution, vortexed 30 s every 30 min for six times in total, placed the sample in a refrigerator at 4 °C overnight. Following centrifugation for 10 min, the extracts were filtrated (SCAA-104, 0.22 μm pore size; ANPEL, Shanghai, China, http://www.anpel.com.cn/) before UPLC-MS/MS analysis.

The sample extracts were analyzed using an UPLC-ESI-MS/MS system (UPLC, SHIMADZU Nexera X2, www.shimadzu.com.cn/; MS, Applied Biosystems 4500 Q TRAP, www.appliedbiosystems.com.cn/). The analytical conditions were as follows, UPLC: column, Agilent SB-C18 (1.8 μm, 2.1 mm*100 mm); The mobile phase was consisted of solvent A, pure water with 0.1% formic acid, and solvent B, acetonitrile with 0.1% formic acid. Sample measurements were performed with a gradient program that employed the starting conditions of 95% A, 5% B. Within 9 min, a linear gradient to 5% A, 95% B was programmed, and a composition of 5% A, 95% B was kept for 1 min. Subsequently, a composition of 95% A, 5.0% B was adjusted within 1.1 min and kept for 2.9 min. The flow velocity was set as 0.35 ml per minute; The column oven was set to 40 °C; The injection volume was 4 μl. The effluent was alternatively connected to an ESI-triple quadrupole-linear ion trap (QTRAP)-MS.

LIT and triple quadrupole (QQQ) scans were acquired on a triple quadrupole-linear ion trap mass spectrometer (Q TRAP), AB4500 Q TRAP UPLC/MS/MS System, equipped with an ESI Turbo Ion-Spray interface, operating in positive and negative ion mode, and controlled by Analyst 1.6.3 software (AB Sciex). The ESI source operation parameters were as follows: ion source, turbo spray; source temperature 550 °C; ion spray voltage (IS) 5500 V (positive ion mode)/−4500 V (negative ion mode); ion source gas I (GSI), gas II(GSII), curtain gas (CUR) were set at 50, 60, and 250 psi, respectively; the collision-activated dissociation (CAD) was high. Instrument tuning and mass calibration were performed with 10 and 100 μmol/l polypropylene glycol solutions in QQQ and LIT modes, respectively. QQQ scans were acquired as MRM experiments with collision gas (nitrogen) set to medium. DP and CE for individual MRM transitions was done with further DP and CE optimization. A specific set of MRM transitions were monitored for each period according to the metabolites eluted within this period.

Metabolite characterization was based on self-built database metware database (MWDB), and substance characterization was carried out according to secondary spectrum information. Isotope signals, repeated signals containing $K^+$ ions, $Na^+$ ions, and $NH_4^+$ ions, and repeated signals of fragment ions which are other substances with larger molecular weight were removed during analysis. Metabolite quantification was accomplished by multiple reaction monitoring (MRM) analysis of triple quadrupole mass spectrometry. In MRM mode, the quaternary rod were first screened the precursor ions (parent ions) of the target substance, and excluded the ions corresponding to other molecular weight substances to preliminarily eliminate interference. After the precursor ions were induced to ionize by the collision chamber, they broke to form many fragment ions, and then the fragment ions were filtered by triple quadrupole to select a required characteristic fragment ion, which eliminated the interference of non-target ions and made the quantification more accurate and reproducible. After obtaining the metabolite spectrum analysis data of different samples, the mass spectrum peaks of all substances were integrated by peak area, and the mass spectrum peaks of the same metabolite in different samples were integrated and corrected. Qualitative and quantitative mass spectrometry analysis of metabolites in project samples is based on KEGG compound database, MWDB, and multiple response monitoring (MRM). Metabolite identification is based on the accurate mass of metabolites, isotope distribution and retention time (RT) of MS2 fragments and MS2 fragments. Through the intelligent secondary spectrum matching method independently developed by Metware, Wuhan, the secondary spectrum and RT of metabolites in project samples were intelligently matched with the secondary spectrum and RT of the company database one by one, and the MS tolerance and MS2 tolerance were set to 2 and 5 ppm, respectively.

Unsupervised principal component analysis (PCA) was performed by statistics function within R base package (version 3.5.0) (www.r-project.org). The data was unit variance scaled before unsupervised PCA. The hierarchical cluster analysis (HCA) results of samples and metabolites were presented as heatmaps with dendrograms, while Pearson correlation coefficients (PCC) between samples were calculated in R base package (version 3.5.0) and presented as only heatmaps. Both HCA and PCC were carried out by R package pheatmap (version 1.0.12). For HCA, normalized signal intensities of metabolites (unit variance scaling) were visualized as a color spectrum. Significantly regulated metabolites between groups were determined by VIP ≥ 1 and absolute Log2FC (fold change) ≥ 1. VIP values were extracted from OPLS-DA result, which also contain score plots and permutation plots, was generated using R package MetaboAnalystR (version 1.0.1). The data was log transform (log2) and mean centering before OPLS-DA. In order to avoid overfitting, a permutation test (200 permutations) was performed. In OPLSDA analysis, the original data was transformed by log2, and then processed by Mean Centering, and then analyzed by OPLSR. Identified metabolites were annotated using KEGG Compound database (http://www.kegg.jp/kegg/compound/), annotated metabolites were then mapped to KEGG Pathway database (http://www.kegg.jp/kegg/pathway.html). Pathways with significantly regulated metabolites were mapped and fed into metabolite sets enrichment analysis (MSEA). Their significance was determined by hypergeometric test's p-values.

The metabolites detected by the LC–MS system were divided into two qualitative grades, level A and level B. Level A: The matching rate between the signals were

collected by actual samples and the signals of the local database substance exceeds 90%, and the secondary mass spectrometry and RT of the sample substances were consistent with the database substances. Level B: The matching rate between signals were collected by actual samples and signals of local database substances exceeds 60%, and Q1 (the mother ion (m/z), mass fraction of protonation), Q3 (the sub-ions, generally characteristic fragment ions), RT, declustering potential (DP) and collision energy (CE) of the sample substances were consistent with the database substances. Qualitative and quantitative mass spectrometry analysis of metabolites in project samples was based on KEGG compound database, MWDB and MRM. Metabolite identification was based on the accurate mass of metabolites, MS2 fragments, MS2 fragments isotope distribution and RT. Through the company's self-developed intelligent secondary spectrum matching method, the secondary spectrum and RT of the metabolites in the project samples were compared with the company's The database secondary spectrum and RT were intelligently matched one by one, and the MS tolerance and MS2 tolerance were set to 2 and 5 ppm, respectively.

**Determination of total N concentrations of panicles**. Fresh panicles were sampled and placed in glass Petri dishes with two layers of qualitative filter paper, and then were dried in an oven at 80 °C for 72 h for fixation. To ensure the strict uniformity of the samples and meet the requirements of determination of N concentrations, the samples were loaded into round-bottom centrifugal tubes within steel balls for grinding at a rate of 23 times per second for 50–60 min. The size of the centrifugal tubes were adjusted according to the volume of the sample. During the grinding process, every 3 min of grinding was followed with 3 min of interruption to prevent the samples friction heat. After tissue homogenization, N concentrations were determined using an elemental analyzer (IsoPrime100; Elementar). All experiments were conducted with at least two replicates.

**Phylogenetic tree construction and multiple sequence alignment**. The amino acid sequence of the ETF domain was used as the query to perform PSI-BLAST and SMART (http://smart.embl-heidelberg.de/index2.cgi). Target sequences were downloaded and used for constructing neighbor-joining trees by MEGA5 software. Multiple sequence alignment was conducted using the online toolkit MAFFT (https://toolkit.tuebingen.mpg.de/mafft), and the result was visualized using ESPRIPT3 (http://espript.ibcp.fr/ESPript/ESPript/).

**Reporting summary**. Further information on research design is available in the Nature Research Reporting Summary linked to this article.

## Data availability
The data supporting the findings from this study are available within the article file and its Supplementary Information. The additional details on map-based cloning are provided as Supplementary Data 1. And the raw images of microscopy data and the metabolomics data are provided in the Source Data file. The microscopy data are available at BioStudies with the accession number S-BSST743. The raw metabolomics data have been deposited in Metabolights, and the unique identifier is MTBLS3924. Any remaining raw data will be available from the corresponding author upon reasonable request. Source data are provided with this paper.

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

## Acknowledgements

This work was supported by the Strategic Priority Research Program of Chinese Academy of Sciences (XDA 24010302), and the National Natural Science Foundation of China (grants 31930018 and 31971912).

## Author contributions

H.Y. and Y.L. conceived the research; H.Y. and Y.L. performed most biological experiments; H.Y., W.S., and Y.S. did the informatics analysis; H.Y., Y.C., Y.S., E.X., N.M. analyzed the data; D.T. and G.D. developed plant materials; H.Y. wrote the article; Z.C. supervised and completed the writing.

## Competing interests

The authors declare no competing interests.
