## [Peer Review File · Nature Communications]

N nutrition contributes to plant fertility by affecting meiosis initiationREVIEWER COMMENTS

Reviewer #1 (Remarks to the Author):

This manuscript proved that the rice ETFB, a subunit of electron transfer flavoprotein putatively required for reutilization of nitrogen (N), has some impacts on meiosis.

The authors selected a sterile etfb mutant from gamma-ray irradiated lines. This mutant showed severe defects in development or differentiation of sporogenous cells (SCs), meiocyte precursors. From results of RNA in situ hybridization of marker genes, MEL1 and REC8, meiosis initiation was supposed to be affected by the mutation. Interestingly, the mutant phenotypes were significantly attenuated when the plants were grown at the other place, probably due to differences in nutritional soil conditions. The authors grew the plants in different nutritional conditions, and as expected, the phenotypes were fully recovered in a higher nutritional condition. It is sure that sufficient N supply was depleted in etfb mutant anthers (Fig. 5), and the phenotype is recovered by exogenously supplied N (Fig. 6). Thus the authors concluded that meiosis initiation is N-dependent in rice.

All results are new findings and the authors' hypothesis is attractive, considering the nitrogen starvation triggers meiosis in yeast. Thus I think publishing this paper will be beneficial for a broad range of researchers in the field of plant reproduction. However, I have several major and minor questions or comments as follows;

Major points;

(1) The authors concluded that meiosis initiation is N-dependent in rice, because of etfb mutation reducing the expression of MEL1 and REC8 marker genes in young anthers (Fig. 1e). However, MEL1 gene expression begins at the archesporial cell initiation stage, extremely prior to SC development and meiosis initiation. This fact suggests that the ETFB effect is not limited to meiosis initiation, rather it may be on SC development or differentiation. If aberrant SC development is a primary effect of etfb mutation, it is no wonder that meiosis initiation is severely affected. This idea seems supported by observation of anther sections in Fig. 1c. In this context, I don't think the results fully support the title and authors' conclusion.

(2) This manuscript clearly indicates that the ETF/ETFQO system plays an important role in SC development and meiosis in both male and female organs in rice. However, there is a big gap between meiotic events and ETF functions/N stress, and I felt the analyses of this study being still on the way to revealing the mechanisms inducing meiosis.

Minor points;

(1) L4-5, p. 8,

I understand etfb mutant showed more severe abnormalities when grown in Hainan than in Beijing. However, I couldn't understand why the authors concluded it attributable to "early meiosis", but not to premeiosis or SC developmental stages.

(2) The numbers of figures and figure panels irregularly appears in the text. For example, Fig. 7a (L2 from the bottom of p. 6) appears just following Fig. 1. They should be aligned and numbered in the order of their appearance in the text.

(3) L2, p.26,

"Metabonomic"  "Metabolomic"

Reviewer #2 (Remarks to the Author):

The manuscript by Zhukan Cheng and coworkers provides a fascinating insight into the effect of N nutrition in plant fertility by determining meiosis initiation. My own expertise is in plant physiology and metabolism and I will hence restrict myself to these aspects of the manuscript.

I feel that the advance made in these areas is massive. Firstly they provide a mechanistic link to underpin the effect of N nutrition on plant reproductive biology. Secondly, they provide a clear role for the etf complex during normal plant life. As yet its function has only been unveiled under stress conditions. The conclusions and working model drawn from these experiments are clearly and succinctly written and the data compellingly supports these. I feel that the data provided is utterly sufficient to support the conclusions drawn and only have the minor suggestion that the novelty of the study be better stressed. As it is the authors underplay this a little which they should be discouraged from.

The manuscript language is also superb as are the figures presented. I am thus in the rare position of suggesting that the manuscript is essentially accepted as it was submitted.

Response to reviewer comments

Reviewer #1:

This manuscript proved that the rice ETF β , a subunit of electron transfer flavoprotein putatively required for reutilization of nitrogen (N), has some impacts on meiosis.

The authors selected a sterile *etf β* mutant from gamma-ray irradiated lines. This mutant showed severe defects in development or differentiation of sporogenous cells (SCs), meiocyte precursors. From results of RNA in situ hybridization of marker genes, *MEL1* and *REC8*, meiosis initiation was supposed to be affected by the mutation. Interestingly, the mutant phenotypes were significantly attenuated when the plants were grown at the other place, probably due to differences in nutritional soil conditions. The authors grew the plants in different nutritional conditions, and as expected, the phenotypes were fully recovered in a higher nutritional condition. It is sure that sufficient N supply was depleted in *etf β* mutant anthers (Fig. 5), and the phenotype is recovered by exogenously supplied N (Fig. 6). Thus, the authors concluded that meiosis initiation is N-dependent in rice.

All results are new findings and the authors' hypothesis is attractive, considering the nitrogen starvation triggers meiosis in yeast. Thus, I think publishing this paper will be beneficial for a broad range of researchers in the field of plant reproduction. However, I have several major and minor questions or comments as follows;

Major points:

(1) The authors concluded that meiosis initiation is N-dependent in rice, because of *etf β* mutation reducing the expression of *MEL1* and *REC8* marker

genes in young anthers (Fig. 1e). However, *MEL1* gene expression begins at the archesporial cell initiation stage, extremely prior to SC development and meiosis initiation. This fact suggests that the *ETFβ* effect is not limited to meiosis initiation, rather it may be on SC development or differentiation. If aberrant SC development is a primary effect of *ETFβ* mutation, it is no wonder that meiosis initiation is severely affected. This idea seems supported by observation of anther sections in Fig. 1c. In this context, I don't think the results fully support the title and authors' conclusion.

Response: Thank you very much for your careful reviewing and valuable suggestions that our data could not completely rule out the possibility that there are defects in earlier developmental processes including archesporial cells (ARs) and sporogenous cells (SCs) development in *etfβ* (N-free).

According to your suggestion, we added *in situ* expression analysis of *MEL1* in WT, *etfβ* (N-free) and *etfβ* (2N) anthers in ARs (one-layer stage) and primary sporogenous cells (PSCs) (two-layer stage) (Supplementary Fig. 7).

In WT anthers under the N-free condition, a faint *MEL1* mRNA signal was first detected in ARs (one-layer stage). Thereafter the signal became stronger in PSCs (two-layer stage) and SCs (three-layer stage), and was persistently strong in the early pollen mother cells (PMCs). In anthers of *etfβ* (N-free), the *MEL1* mRNA signal was the same as that in WT and *etfβ* (2N) in ARs (one-layer stage) and PSCs (two-layer stage). However, the disparity of *MEL1* mRNA signal in *etfβ* (N-free) compared with WT was first detected in SC-like cells (SCLs) at three-layer stage. The *MEL1* mRNA signal in SCLs of *etfβ* (N-free) was so weak and faded so quickly that it was almost undetectable in SC progenies (SCPs) until the four-layer stage (Fig. 4b).

Additionally, there are no significant cytological abnormalities in *etfβ* (N-free) anthers at one-layer stage and two-layer stage via TBO-and DAPI-stained

another section (Fig. 4 and Supplementary Fig. 5), we speculated that the cytological defects in *effβ* (N-free) was first detected between three- and four-layer stage. And the development and differentiation from three- to four-layer stage is the crucial for meiosis initiation, we therefore hypothesized that the abnormal expression patterns of *MEL1* and *REC8* in SCLs of *effβ* (N-free) were responsible for the failure of meiosis initiation and the generation of SCPs without meiotic fate.

Meanwhile, besides the differentiation of SCLs to PMCs, both the development of SCLs may also be impaired by the abnormal expression pattern of *MEL1* in *effβ* (N-free). However, effective detection methods to determine the developmental defects of SCs are still lacking, therefore we cautiously drew a conservative conclusion that N nutrition contributes to plant fertility by determining meiosis initiation. And we have made corresponding revision and supplement in the title, main text (L18-19, P12; L1-11, P 14; L18-21, P14) and discussion (L3, P 21) in this revised manuscript according to your suggestions.

We hope our revision and supplement in this revised manuscript has made our interpretation more reasonable and logical. Thank you very much.

(2) This manuscript clearly indicates that the ETF/ETFQO system plays an important role in SC development and meiosis in both male and female organs in rice. However, there is a big gap between meiotic events and ETF functions/N stress, and I felt the analyses of this study being still on the way to revealing the mechanisms inducing meiosis.

Response: We totally agree with you on that building more direct and concrete correlations between meiotic events and ETF functions/N stress is essential and significant for us. As you suggested, revealing the dependence of meiosis initiation on N nutrient in rice at molecular and biochemical levels will be a key research direction to which we will devote a lot of time and energy. At present, the mechanism of N nutrition regulating meiosis initiation is relatively clear and

representative in budding yeast (*Saccharomyces cerevisiae*), and other species are still to be explored. Due to the lack of research clues in this area, it is challenging and difficult for us to narrow this big gap, but there is no doubt that we will try our best.

In budding yeast, *Initiator of Meiosis 1 (IME1)* behaves as a key positive regulator of meiosis¹, which is a master regulator to activate the expression of meiotic genes and initiate early meiotic events during the acquisition of meiotic fate². There are four Upstream Control Sequences (UCS) in *IME1* promoter³, and the UCS1 transmits the signal of nitrogen abundance which represses *IME1* transcription⁴. Consequently, N stress can induce *IME1* and trigger meiosis initiation in budding yeast.

As we well known, budding yeast is a single-celled eukaryote whose N utilization and metabolic systems⁵ can be are rather different from rice⁶⁻⁸ (*Oryza sativa*), so do the regulatory mechanisms between N nutrition and meiosis initiation. Even so, the discoveries in budding yeast are valuable for us to learn, and we conservatively speculate that certain similar transcriptional regulation mechanisms might also exist in rice. Given the meiosis initiation defects and endogenous N deficiency in *etfβ*, we have a reasonable conjecture. There may be a transcription factor whose expression level fluctuates with contents of certain cellular nitrogenous substances in rice. Meanwhile, the transcription factor can activate the expression of downstream meiosis initiation genes. If the supposed transcription factor exists as expected, its expression will be downregulated due to the endogenous N deficiency in *etfβ* (N-free), resulting in the meiosis initiation defects.

Consequently, the comparative analysis of RNA-Seq data of (5-cm (± 1 cm) panicles) between *etfβ* (N-free) and *etfβ* (2N) will be significant for us to search for the supposed transcription factor. Then the differently expressed genes should be compared with RNA-Seq data of the meiotic entrance mutant under

2N (*sp^l* can be a candidate) whose defects are inherent to filter out known meiosis genes. To verify whether the function of this transcription factor meets our expectations, knocking out it in WT and overexpressing it in *etfβ* will be necessary. The knockout mutant cannot enter meiosis regardless of the exogenous N level, and the phenotypes of *OE/ etfβ* (N-free) can be rescued. If both conditions are satisfied, the transcription factor may be the candidate. We hope our experimental designs and corresponding explanations have positive significance for bridging the gap between meiotic events and ETF functions/N stress.

So far, we have provided compelling evidence that N nutrition contributes to plant fertility by affecting meiosis initiation, but the deeper metabolic and biochemical functions of electron transfer flavoprotein (ETF)/ electron transfer flavoprotein quinone oxidoreductase (ETFQO) system remain to be explored. Thus, we obtained knockout mutant of other four genes in ETF/ETFQO system, *isovalyl coenzyme A dehydrogenase (IVDH)*, *D-2-hydroxyglutarate dehydrogenase (D2HGDH)*, *ETF subunit α (ETFα)* and *ETFQO* via CRISPR-CAS9 system. The nutrient treatment experiments for these mutants have been done, and follow-up experiments are under way. We will do our best to make further progress in revealing the function of ETF/ETFQO system in gametogenesis at molecular and biochemical levels.

Minor points:

(1) L4-5, p. 8, I understand *etfβ* mutant showed more severe abnormalities when grown in Hainan than in Beijing. However, I couldn't understand why the authors concluded it attributable to "early meiosis", but not to premeiosis or SC developmental stages.

Response: Thanks for your careful reviewing. We agree with you opinion that it is not rigorous to speculate that ETFβ is involved in early meiosis according

to the phenotypic comparison of mutant anther from spikelets of 6mm between Beijing and Hainan (Fig. 1c, d), as the possibility that premeiosis and SC development in *effβ* is defected cannot be ruled out here. The conjecture we made here is indeed imprecise, because the possibility of abnormal SC development was determined according to transverse section of mutant anther from one layer stage to mature pollen stage (Fig. 4a). We revised that statement of the assumption (L10-11, P 7; L5-6, P 8) and hope the modifications make it more logical and clearer. Thanks a lot.

(2) The numbers of figures and figure panels irregularly appears in the text. For example, Fig. 7a (L2 from the bottom of p. 6) appears just following Fig. 1. They should be aligned and numbered in the order of their appearance in the text.

Response: Thanks for your kind reminding. We have extracted the data supporting the conclusion here from Fig. 7a, and reintegrated it into the Supplemental Fig. 1 as Supplemental Fig. 1c. Moreover, we have relabeled other figure information to align figures regularly (L20, P. 6; L1-5, P 7; L7, P 7; L14, P 7; L2, P 7; L7, P 16; L20, P 16; L19, P 17). Thanks a lot.

(3) L2, p.26,

"Metabonomic"  "Metabolomic"

Response: Thank you for your careful reviewing. We have corrected the spelling mistake as you indicated (L13, P.26). We should apologize for this mistake. Thanks a lot.

Reviewer #2 (Remarks to the Author):

The manuscript by Zhukan Cheng and coworkers provides a fascinating insight into the effect of N nutrition in plant fertility by determining meiosis initiation. My

own expertise is in plant physiology and metabolism and I will hence restrict myself to these aspects of the manuscript.

I feel that the advance made in these areas is massive. Firstly, they provide a mechanistic link to underpin the effect of N nutrition on plant reproductive biology. Secondly, they provide a clear role for the ETF complex during normal plant life. As yet its function has only been unveiled under stress conditions. The conclusions and working model drawn from these experiments are clearly and succinctly written and the data compellingly supports these. I feel that the data provided is utterly sufficient to support the conclusions drawn and only have the minor suggestion that the novelty of the study be better stressed. As it is the authors underplay this a little which they should be discouraged from.

The manuscript language is also superb as are the figures presented. I am thus in the rare position of suggesting that the manuscript is essentially accepted as it was submitted.

Response: Thanks sincerely for your careful reviewing and positive encouragement and appreciation. We would like do our best to make further progress in revealing the function of electron transfer flavoprotein (ETF)/electron transfer flavoprotein quinone oxidoreductase (ETFQO) system in gametogenesis at molecular and biochemical levels. So far, we have done the nutrient treatment experiments for the mutants of other four genes in ETF/ETFQO system, *isovalyl coenzyme A dehydrogenase (IVDH)*, *D-2-hydroxyglutarate dehydrogenase (D2HGDH)*, *ETF subunit α (ETF α)* and *ETFQO*, and follow-up explorations are under way. We would try our best to make further progress in revealing the function of ETF/ETFQO system in gametogenesis at molecular and biochemical levels. Thank you very much.

References

- 1 Granot, D., Margolskee, J. P. & Simchen, G. A long region upstream of the IME1 gene regulates meiosis in yeast. *Mol Gen Genet* **218**, 308-314, doi:10.1007/BF00331283 (1989).
- 2 Kassir, Y., Granot, D. & Simchen, G. IME1, a positive regulator gene of meiosis in *S. cerevisiae*. *Cell* **52**, 853-862, doi:10.1016/0092-8674(88)90427-8 (1988).
- 3 Piekarska, I., Rytka, J. & Rempola, B. Regulation of sporulation in the yeast *Saccharomyces cerevisiae*. *Acta Biochim Pol* **57**, 241-250 (2010).
- 4 Wannige, C. T., Kulasiri, D. & Samarasinghe, S. A nutrient dependant switch explains mutually exclusive existence of meiosis and mitosis initiation in budding yeast. *J Theor Biol* **341**, 88-101, doi:10.1016/j.jtbi.2013.09.030 (2014).
- 5 Tudzynski, B. Nitrogen regulation of fungal secondary metabolism in fungi. *Front Microbiol* **5**, doi:ARTN 65610.3389/fmicb.2014.00656 (2014).
- 6 Chardon, F., Noel, V. & Masclaux-Daubresse, C. Exploring NUE in crops and in *Arabidopsis* ideotypes to improve yield and seed quality. *Journal of Experimental Botany* **63**, 3401-3412, doi:10.1093/jxb/err353 (2012).
- 7 Luo, L., Zhang, Y. L. & Xu, G. H. How does nitrogen shape plant architecture? *Journal of Experimental Botany* **71**, 4415-4427, doi:10.1093/jxb/eraa187 (2020).
- 8 Li, H., Hu, B. & Chu, C. Nitrogen use efficiency in crops: lessons from *Arabidopsis* and rice. *J Exp Bot* **68**, 2477-2488, doi:10.1093/jxb/erx101 (2017).
- 9 Ren, L. *et al.* OsSPL regulates meiotic fate acquisition in rice. *New Phytol* **218**, 789-803, doi:10.1111/nph.15017 (2018).